# Understanding the Antilymphoma Activity of *Annona macroprophyllata* Donn and Its Acyclic Terpenoids: In Vivo, In Vitro, and In Silico Studies

**DOI:** 10.3390/molecules27207123

**Published:** 2022-10-21

**Authors:** Jesica Ramírez-Santos, Fernando Calzada, Jessica Elena Mendieta-Wejebe, Rosa María Ordoñez-Razo, Rubria Marlen Martinez-Casares, Miguel Valdes

**Affiliations:** 1Instituto Politécnico Nacional, Escuela Superior de Medicina, Sección de Estudios de Posgrado e Investigación, Plan de San Luis y Salvador Díaz Mirón S/N, Col. Casco de Santo Tomás, Mexico City 11340, Mexico; 2Unidad de Investigación Médica en Farmacología, UMAE Hospital de Especialidades 2° Piso CORSE Centro Médico Nacional Siglo XXI, Instituto Mexicano del Seguro Social, Av. Cuauhtémoc 330, Col. Doctores, Mexico City 06720, Mexico; 3Unidad de Investigación Médica en Genética Humana, UMAE Hospital Pediatría, 2° Piso, Centro Médico Nacional Siglo XXI, Instituto Mexicano del Seguro Social, Av. Cuauhtémoc 330, Col. Doctores, Mexico City 06725, Mexico; 4Departamento de Sistemas Biológicos, Universidad Autónoma Metropolitana, Calzada del Hueso 1100, Mexico City 04960, Mexico

**Keywords:** geranylgeraniol, farnesyl acetate, phytol, cytotoxic activity, antilymphoma activity, U-937 cells, brine shrimp lethality, acute toxicity, GC-MS analysis

## Abstract

*Annona macroprophyllata* Donn (*A. macroprophyllata*) is used in traditional Mexican medicine for the treatment of cancer, diabetes, inflammation, and pain. In this work, we evaluated the antitumor activity of three acyclic terpenoids obtained from *A. macroprophyllata* to assess their potential as antilymphoma agents. We identified the terpenoids farnesyl acetate (FA), phytol (PT) and geranylgeraniol (Gg) using gas chromatography–mass spectroscopy (GC-MS) and spectroscopic (^1^H, and ^13^C NMR) methods applied to petroleum ether extract of leaves from *A. macroprophyllata* (PEAm). We investigated antitumor potential in Balb/c mice inoculated with U-937 cells by assessing brine shrimp lethality (BSL), and cytotoxic activity in these cells. In addition, to assess the potential toxicity of PEAm, FA, PT and Gg in humans, we tested their acute oral toxicity in mice. Our results showed that the three terpenoids exhibited considerable antilymphoma and cytotoxic activity. In terms of lethality, we determined a median lethal dose (LD_50_) for thirteen isolated products of PEAm. Gg, PT and AF all exhibited a higher lethality with values of 1.41 ± 0.42, 3.03 ± 0.33 and 5.82 ± 0.58 µg mL^−1^, respectively. To assess cytotoxic activity against U-937 cells, we calculated the mean cytotoxic concentration (CC_50_) and found that FA and PT were closer in respect to the control drug methotrexate (MTX, 0.243 ± 0.007 µM). In terms of antilymphoma activity, we found that FA, PT and Gg considerably inhibited lymph node growth, with median effective doses (ED_50_) of 5.89 ± 0.39, 6.71 ± 0.31 and 7.22 ± 0.51 mg kg^−1^ in females and 5.09 ± 0.66, 5.83 ± 0.50 and 6.98 ± 0.57mg kg ^−1^ in males, respectively. Regarding acute oral toxicity, we classified all three terpenoids as category IV, indicating a high safety margin for human administration. Finally, in a molecular docking study of 3-hydroxy-3-methylglutaryl-coenzyme A (HMG-CoA) reductase, we found binding of terpenoids to some amino acids of the catalytic site, suggesting an effect upon activity with a resulting decrease in the synthesis of intermediates involved in the prenylation of proteins involved in cancer progression. Our findings suggest that the acyclic terpenoids FA, PT, and Gg may serve as scaffolds for the development of new treatments for non-Hodgkin’s lymphoma.

## 1. Introduction

*Annona macroprophyllata* Donn (*A. macroprophyllata*), of the family Annonaceae, is used in Mexican traditional medicine for the treatment of cancer, diabetes, inflammation and pain [1,2]. In Central America, it is commonly known as “ilama”, “papauce”, “cherimoya” or “lamazapotl” [3]. It is a species which thrives in tropical climates, and its exotic fruits have a commercial value [4]. *A. macroprophyllata* contains a wide variety of secondary metabolites, such as alkaloids, flavonoids, acetogenins and terpenoids [5,6,7,8]. Ethanol and hexanic extracts made from its leaves and seeds exhibit pharmacological properties including anticonvulsant, antifungal, antinociceptive, antimicrobial, antihyperglycemic and antitumor effects, as well as inhibition of the α-glucosidase enzyme [9,10,11,12].

Terpenoids are present in a wide variety of plants and animals. They form the largest group of secondary metabolites in plants and play an important role in ecological and biological processes such as repelling or attracting of insects; they are also responsible for the fragrance of many plants [13]. Today, there is increasing interest in obtaining these molecules for commercial and pharmaceutical use due to their cytotoxic, antimicrobial, antifungal, antiviral, antihyperglycemic, antitumor, antiparasitic and anti-inflammatory properties [14]. 

Researchers have found chemopreventive effects of terpenoids in breast, liver, skin, lung, colon, and prostate cancer associated with their ability to regulate several intracellular signaling pathways which act in early and late stages of cancer progression by inducing apoptosis and cell cycle arrest, and suppressing cell differentiation, angiogenesis, invasion and metastasis [14,15].

Non-Hodgkin’s lymphoma (NHL) is a type of cancer which manifests as lymphadenopathies or solid tumors affecting the lymphatic organs and blood precursor cells, mainly B and T cells, with more than fifty classified subtypes, according to the World Health Organization [16]. In 2020, NHL caused the death of 259,793 individuals worldwide [17]. In Mexico, statistical data of incidence and mortality for the year 2020 show that this disease was amongst the top four causes of death in populations of both sexes aged from 1 to 29 years [18], with 6840 new cases and 3071 deaths reported [19]. Some of the most common medications used alone or in combination for the treatment of NHL are cyclophosphamide, doxorubicin, vincristine, prednisone, cisplatin, fludarabine, methotrexate (MTX) and etoposide. However, all these medications have several short- and long-term side effects and limited therapeutic effects [16,20]. Short-term side effects include myelosuppression, thrombocytopenia, neutropenia and anemia [16,20]. Long-term side effects include cardiomyopathy, peripheral neuropathy, hypogammaglobulinemia, acute kidney injury, genotoxic damage, and hepatotoxicity [16,20]. Given this context, there is an obvious need for new antitumor agents with fewer side effects and increased therapeutic efficacy.

In a previous work, we evaluated the preventive antilymphoma effects of petroleum ether extract obtained from *A. macroprophyllata* (PEAm) leaves, and of geranylgeraniol (Gg), a compound isolated from PEAm [2]. To continue the search for new antitumor agents, we sought in this study to explore the therapeutic activity of PEAm, Gg, farnesyl acetate (FA) and phytol (PT) by means of activity-guided isolation and identification methods, using Balb/c mice with induced lymphoma. We assessed lethality using the brine shrimp lethality test (BSL) and by measuring cytotoxic effects against U-937 cells. We determined safety margins using the acute oral toxicity test. Finally, we carried out a molecular docking study on a possible target to further investigate the antilymphoma effects of Gg, FA, and PT.

## 2. Results

### 2.1. Brine Shrimp Lethality Assay

Results of our brine shrimp acute lethality test (Table 1) showed that PEAm and its thirteen products isolated by preparative thin-layer chromatography (TLC) were lethal to *Artemia salina* larvae. All products exhibited greater activity than MTX with a median lethal concentration (LD_50_) value of <100 µg mL^−1^. The products Gg (P6), PT (P7) and FA (P10) were most active. These exhibited LD_50_ values of <10 µg mL^−1^ with Gg the most active of all (LD_50_ = 1.41 ± 0.42). We also found that PEAm (18.84 ± 0.58 µg mL^−1^) was more active than MTX (24.66 ± 0.27 µg mL^−1^).

After recording the lethality of each product, we selected those compounds with the lowest LD_50_ values and selected Gg (P6), PT (P7) and FA (P10) for identification and characterization, as indicated below.

### 2.2. Analysis of GC-MS, and ^1^H-^13^C-NMR Spectra of Geranylgeraniol (P6), Phytol (P7) and Farnesyl Acetate (P10)

We performed the PEAm analysis by GC-MS and used standards for the terpenoids farnesyl acetate, phytol, and geranylgeraniol (Figure 1). The analysis showed the presence of high levels of farnesyl acetate at 17.95 min and lower concentrations of phytol at 23.07 min and geranylgeraniol at 24.12 min (Figure 2 and Table 2). The identification was performed using their mass spectra (Figure 3) in comparison with the NIST mass spectra libraries. Furthermore, we obtained the same retention factor when performing thin layer chromatography and comparing compounds P6, P7 and P10 with authentic sigma standards. To confirm the identification of farnesyl acetate, phytol and geranylgeraniol, we characterized by spectroscopic (^1^H, and ^13^C NMR spectra) methods (Table 3) (Figure 4 and Figure 5) which were identical with reported in the literature.

### 2.3. Citotoxic Activity

The cytotoxic activity against U-937 cells (Table 4) showed that farnesyl acetate (FA, CC_50_ 0.275 ± 0.08 µM) and phytol (PT, 0.296 ± 0.07 µM) were closer in respect to the control drug methotrexate (MTX, 0.243 ± 0.007 µM). Furthermore, the effect of geranylgeraniol (Gg, 0.395 ± 0.005 µM) was lower with the cytotoxicity of MTX. PEAm was three-fold less active than MTX. In general, PEAm and its acyclic terpenoids showed a dose-dependent cytotoxic activity.

### 2.4. Antilymphoma Activity

At day 28, taking as reference the clinical staging of NHL, we classified all mice as being at least stage III, because we found considerable differences in weight gain of the axillary and inguinal nodes for animals of both sexes, compared with the healthy control (HC) group (Figure 6). After we administered treatments, we found that FA produced greater antilymphoma activity in female mice (ED_50_ 5.89 ± 0.39 mg/kg) and in male mice (ED_50_ 5.09 ± 0.66 mg/kg). We also found that PT produced a good effect in male mice (ED_50_ 5.83 ±0.50 mg/kg) while PEAm produced a greater effect in male mice (166.41 ± 3.8 mg/kg) (Table 5). In general, we found that petroleum ether extract from *A. macroprophyllata* leaves (PEAm) and its acyclic terpenoids produced substantial dose-dependent antilymphoma activity in female mice and non-dose-dependent activity in male mice. 

### 2.5. Acute Oral Toxicity

The results of acute oral toxicity revealed that PEAm and its isolated terpenoids FA and PT did not cause behavioral changes and did not cause mortality in the animals until the end of the study. At day 14, macroscopic observation of the internal organs revealed no tissue damage or weight loss. PEAm was classified within category 5 (LD_50_ > 3000 mg kg^−1^), which suggests that its use is safe in humans. In the case of FA and PT, their classification was category 4 (LD_50_ > 1000 mg kg^−1^) and therefore low risk. In the case of Gg and MTX, they were also classified in category 4. In the macroscopic study, no tissue lesions or loss or gain in weight of the internal organs were observed. However, they caused death in the animals, obtaining an LD_50_ of 742.17 ± 0.34 and 335.04 ± 0.39 mg kg^−1^, respectively (Table 4). These results and the results of the antilymphoma activity allowed us to obtain the therapeutic index (TI) for each evaluated compound. The results did not reveal significant changes in the increase or decrease in the TI values of each molecule when comparing the results in both sexes; however, the female mice presented slightly lower values compared to the values obtained in males. The TI for FA, PT and Gg was >100 in both sexes, which suggests that these molecules have a high safety margin for their use. In the case of MTX, the TI results were >200 for both sexes (Table 6).

### 2.6. Molecular Docking Studies of Geranylgeraniol (Gg), Farnesyl Acetate (FA), and Phytol (PT)

As part of the ongoing quest for therapeutic targets in cancer, we carried out a molecular docking using the HMG-CoA reductase enzyme as a target, in the knowledge that this is the regulatory enzyme of the mevalonate pathway. This pathway contributes to the modulation of various pathways involved in cancer progression through the prenylation of RAS, RAC, Rho and GTPases proteins related to cell growth and cell proliferation in cancer [21,22]. In addition, terpenoids can bring about an impact on the enzyme HMG-CoA reductase [14,23,24].

The results of the molecular docking study on HMG-CoA reductase are shown in Figure 7 and Table 7. FA, PT and Gg showed a binding energy of −7.38, −7.55, −7.95 kcal-mol^−1^, respectively. The HMG-CoA-substrate showed the highest binding energy with −9.21 kcal-mol^−1^. In addition, FA, PT, Gg and HMG-CoA-substrate obtained the best possesses with the similar shared polar interaction in the cis-loop in Asn 755 on HMG-CoA reductase; PT and Gg share other polar interactions at Lys 691 and Leu 562, respectively. The three acyclic terpenoids showed to share several polar interactions in the catalytic portion of the enzyme (residues 426-888), such as Thr 557, Glu 559, Thr 558, Thr 758, Ile 762, Gly 765, Gln 766, Asp 767, Gly 807, Gly 808. These results could suggest the blockade of the access of the HMG-CoA substrate and confirm the role of terpenoids on the enzyme HMG-CoA reductase.

## 3. Discussion

Cancer is one of the leading causes of death around the world. In the year 2020, non-Hodgkin’s lymphomas ranked eleventh in terms of mortality worldwide and fourth in Mexico [17,18]. Typically, initial treatment for NHL involves chemotherapy, either alone or in combination with other interventions. Such treatments are associated with high cure rates in most NHL cases, especially when the patient is diagnosed in an early stage of the disease or presents with a lower grade of lymphoma. However, chemotherapy also produces several short- and long-term side effects that substantially decrease the patient’s quality of life. In addition, chemotherapy using current management strategies has limited beneficial effects upon many NHL subtypes [16,20].

The search for agents with possible antilymphoma effects continues [2]. In this study, we sought to assess the antilymphoma activity of PEAm and the three isolated terpenoids farnesyl acetate (FA), phytol (PT) and geranylgeraniol (Gg) over a 65-day modeling period. We strategically evaluated their therapeutic effects using activity-guided isolation. We also assessed their cytotoxicity and acute oral toxicity effects.

Firstly, we established an in vivo model using the U-937 cell line. We used this cell line because it is more aggressive, has a histiocytic subtype, and exhibits diffuse proliferation; moreover, it is routinely used for the evaluation of antitumor and cytotoxic activity in natural products [25,26,27,28]. At day 29, we found substantial differences in the weight gains of the left and right inguinal and axillary lymph nodes, compared to our healthy control (Figure 6). We consulted the clinical staging classification described in the Ann Arbor staging system for lymphoma [16,20] and found that the mice in our model with induced lymphoma at day 29 could be classified within stage III, because the growth of the lymph nodes occurred in the area above and below the diaphragm in animals of both sexes. We then investigated the antilymphoma effects of PEAm on the in vivo model. Our findings confirmed this antilymphoma activity, with ED_50_ values of 180.42 mg kg^−1^ in female mice and 166.41 mg kg^−1^ in male mice, in line with the results of our previous study [2], and also those of the work group [25] which investigated the antilymphoma activity of the ethanolic extract obtained from *Annona muricata* leaves and obtained an ED_50_ value of 368.5 mg kg^−1^ for male mice. In another study of the antitumor and antiproliferative activity of *A. macroprophyllata*, the authors of [10] isolated two acetogenins obtained from a hexanic extract of seeds evaluated in the HeLA and SW480 cell lines. Data obtained from studies of other species belonging to the same genus provide further support for the antilymphoma effect of PEAm [25,29,30].

The next step in our study involved isolation and purification of the products by preparative thin-layer chromatography starting from PEAm. By this method, we obtained thirteen products which we then subjected to lethality evaluation by means of an BSL assay which has been previously used as a bioindicator in the screening of possible molecules with antitumor potential [2,31,32]. Our results showed that Gg (P6), PT (P7) and FA (P10) induced a higher dose-dependent lethality with LD_50_ values of <10 µg mL^−1^, representing a lethality greater than that of MTX (LD_50_ = 24.66 ± 0.27 µg mL^−1^) (Table 1). In line with the findings of previous studies, our results suggested that these molecules might have antitumor potential [31,32,33], so we carried out a further investigation of P6, P7 and P10. Later, we characterized P6, P7 and P10 by GC-MS analysis (Table 2, Figure 2 and Figure 3), focusing on volatile compounds of a terpene nature, since previous studies have reported the presence of acyclic terpenoids in *A. macroprophyllata* [34] with antitumor activity [2]. Afterwards, we then obtained an equal retention factor for each compound (P6, P7 and P10) by thin layer chromatography against their authentic sample (Sigma-Aldrich) and revealed them with 10% H_2_SO_4_ (data not shown here). Furthermore, we structurally characterized P6, P7 and P10 by ^1^H and ^13^C NMR spectroscopic methods (Table 3). We identified product P10 was as the sesquiterpene farnesyl acetate (FA), P7 as the sesquiterpene phytol (PT) and P6 as the diterpene geranylgeraniol (Gg) (Figure 4 and Figure 5). These results confirmed the identification of geranylgeraniol in the PEAm, as reported in our previous work [2]. To the best of our knowledge, FA and PT have not been previously identified in this plant species, although previous published works have reported the presence of flavonoids, alkaloids and acetogenins, among other metabolites [5,6,7,8,9,10,11,12,35].

After we identified the terpenoids, we evaluated their cytotoxic activity against the U-937 cell line using the WST-1 assay (Table 4). We found that PEAm and the three terpenoids isolated exhibited high dose-dependent antiproliferative activity against U-937 after 24 h. FA and PT exhibited a closer cytotoxic effect (CC_50_ 0.275 ± 0.08 µM, CC_50_ 0.296 ± 0.07 µM) compared with the reference drug, MTX (CC_50_ 0.243 ± 0.007 µM). In addition, the cytotoxicity against the U-937 cell line obtained from MTX is lower compared to the results of the work carried out through a viability analysis by flow cytometry with a CC_50_ 0.5 µM [36], the cytotoxic effect of FA, PT and Gg is consistent with respect to other studies, since the cytotoxic activity of other terpenoids on some cancer cell lines reported a range of 0.1–100 µM [13,37,38,39]. These results are consistent with reports on the antiproliferative activity of some species that make up the genus Annona, including *Annona muricata, Annona vepretorum, Annona leptopetala, Annona sylvatica, Annona pickelii, Annona salzmannii* [37] and *A.*
*macroprophyllata* [10]. With this in mind, we evaluated the antilymphoma activity of PEAm, FA, PT and Gg in male and female Balb/c mice inoculated with cell line U-937. We found that the three acyclic terpenoids produced a greater inhibition of lymph node growth in both sexes, thus exhibiting a greater antilymphoma effect, compared with PEAm (Table 5). The sesquiterpene FA induced a higher antilymphoma activity (female ED_50_ 5.89 ± 0.39 and male ED_50_ 5.09 ± 0.66 mg kg^−1^). Our antilymphoma activity results showed a direct correlation with the in vitro cytotoxic activity of the three terpenoids. In addition, and in line with previous studies, we found a correlation of antilymphoma activity and BSL by obtaining LD_50_ values of <10 µg mL^−1^ for FA and PT, and <100 µg mL^−1^ for PEAm and MTX which were in line with the Gg and PEAm data reported in our previous study [2,32]. These results confirmed that BSL could be used as a practical, economical, and simple tool for obtaining new antitumor agents isolated from plant extracts. We suggest that the greater antilymphoma effect of sesquiterpene FA might be due to the presence of the acetate group in C1, compared with the hydroxyl in PT and Gg diterpenes; however, additional tests evaluating other acyclic terpenoids are needed to fully confirm this effect.

We also used LD_50_ values for PEAm, AF, PT, Gg and MTX to determine the safety of their use by means of an acute oral toxicity test (Table 6). Following guideline 423 of the Organization for Cooperation and Economic Development (OECD), we classified PEAm as category 5 (LD_50_ > 3000 mg kg^−1^) and safe to use; we classified FA and PT terpenoids as category 4 (LD_50_ > 1000 mg kg^−1^) and low risk. These did not produce side effects such as tremors, convulsions, salivation, diarrhea, lethargy, drowsiness, and coma [40]. We also classified Gg and MTX as category 4, although LD_50_ values for these were lower than those for FA and PT. 

With these data obtained, we sought to calculate TI, and thereby determine the safety margins of PEAm, Gg, AF and PT. In general, the three terpenoids exhibited a high safety margin of >100; however, we suspect the TI may be much higher, because we used an LD_50_ value of mg kg^−1^ for calculation purposes, and our results indicate an actual LD_50_ value of >1000 mg kg^−1^. Due to this, we suggest that further tests should be carried out to obtain a more precise determination of TI. Furthermore, although MTX exhibited the highest TI, it is a drug with serious side effects, depending on duration of treatment, and patient age and condition [41]. For this reason, although MTX is regularly used alone or in combined regimens to treat various types of cancer, including hematological malignancies, it is critical to monitor dosage and assess possible side effects in patients.

Previous findings offer further support for the antineoplastic role found in our work. Researchers have suggested that terpenoids may have potential as chemopreventive and therapeutic agents in various types of cancer including liver, mammary, skin, colon, lung, prostate and pancreatic carcinomas [14,15,35,38,39]. The chemopreventive and therapeutic activity of these molecules is achieved by their acting on several signaling pathways that play an important role during the initiation phase of carcinogenesis, by preventing the interaction of carcinogens with DNA, and also in the promotion phase, by generating cell cycle arrest, preventing damage to DNA and inducing apoptosis [13,14,37,39,42,43]. Our findings of antilymphoma activity of terpenes isolated from PEAm are consistent with previously published reports. Specifically, the authors of [44] found that sesquiterpene FA induced cytotoxicity against the HL-60 cell line [44]. In addition, researchers have found that Gg exhibits important chemopreventive and suppressive ac-tivities on cancer cell lines by arresting the cell cycle in the G1 phase and inducing the down-expression of cyclin D1 and induced apoptosis by activating caspases 8, 9, and 3 in cell lines DU145 and DLD1 [45,46,47,48]. Gg also inhibits the activation of nuclear factor kappa beta, a transcription factor that is aberrantly activated in various cancer cell lines [45]. Researchers have also found antiproliferative activity of PT on the NCCIT cell line, including apoptosis in A549 cells through a depolarization of the mitochondrial membrane potential [22]. Taken together, these findings provide further support for the study of the antitumor activity of terpenoids as agents and thereby obtain new molecules with potential value in cancer chemoprevention and therapy.

Finally, we carried out a molecular docking study. We chose the enzyme HMG-Co A reductase as a target in light of previous studies reporting an increased presence of this enzyme in various types of cancer [49,50], as well as the inhibition of terpenoid-mediated mevalonate synthesis with application to cancer chemotherapy [42,51,52]. Our results showed that FA, PT and Gg showed similar binding energy; however, Gg showed the best interaction on HMG-CoA reductase with a binding energy of −7.95 kcal-mol^−1^ with HMG-CoA-substrate exhibiting a maximum binding energy of −9.21 kcal-mol−1. FA, Gg and PT shared binding at Asn 755, an important amino acid in the catalytic site of the enzyme PT, and Gg also shared important polar interactions at Lys 691 and Leu 562, respectively (Table 7 and Figure 7). In addition, the three molecules studied exhibited several polar interactions within the portion of the enzyme described as the catalytic site. We recall that certain amino acids share polar interactions with some inhibitors known as statins are Glu 559 and Ile 762 [53]. With this in mind, we suggest that the binding of terpenoids to some amino acids of the catalytic site blocks the binding of the HMG-CoA substrate, affects its activity, and results in a decrease in the synthesis of intermediates involved in the prenylation of proteins involved in cancer progression. Acyclic terpenoids might play a role in these phenomena, because previous studies have found that geranylgeraniol plays a role in the negative regulation of the level of the enzyme HMG-CoA reductase, which results in inhibition of the isoprenylation of some proteins in the DU145 cell line, and ultimately leads to apoptosis [46,47,50]. Researchers have also found a decrease in cholesterol levels as an indirect measure of HMG-CoA reductase mediated by Gg [46]. It is well known that the mevalonate pathway provides intermediates such as geranylgeranyl-pyrophosphate and farnesyl-pyrophosphate for the posttranslational prenylation and cell membrane anchoring of several proteins, including the RAS, RAC, Rho, and GTPase families involved in cell growth and cell proliferation in cancer [52,53]; for example, active Rho proteins are involved in numerous cellular events in cancer progression, including cell cycle progression, migration, and transformation [50]. In addition, the authors of [51] found that a decrease in HMG-CoA reductase levels results in the modulation of cell cycle regulation and induction of apoptosis in cancer cells due to a lack of protein prenylation [51].

Our results confirm the antilymphoma activity and cytotoxic properties of three acyclic terpenoids. To the best of our knowledge, this is the first report of antilymphoma activity of FA and PT. We also believe that our identification and isolation of *A. macroprophyllata* are without precedent, as are our findings concerning their activity mediated by the inhibition of the enzyme HMG CoA reductase; however, further tests should be carried out to elucidate and confirm the mechanism by which these molecules act. In summary, our findings confirm the antiproliferative and antitumor activity of terpenoids and highlight the potential importance of these molecules for the development of better antitumor drugs.

## 4. Materials and Methods

### 4.1. Chemicals

Farnesyl acetate (technical, mixture of isomers, PN: 45895-10ML-F), phytol (97%, mixture of isomers, PN: 139912-10G), geranylgeraniol (≥85%, PN: G3278-100MG), methotrexate, dimethyl sulfoxide (DMSO), L-glutamine, penicillin/streptomycin, and RMPI 1640 medium were purchased from Sigma-Aldrich, USA. Bovine fetal serum was purchased from Gibco, Mexico. Petroleum ether, ethyl acetate, and hexane were of analytical grade and were purchased from JT Baker, Mexico.

### 4.2. Extraction and Isolation of Products

We collected *A. macroprophyllata* leaves in Metapa de Dominguez, Chiapas, Mexico (14°50′00″ N 92°11′00″ W). Plant material was identified by M. C. Santiago Xolalpa of the IMSSM Herbarium of the Instituto Mexicano del Seguro Social (IMSS), with voucher specimen corresponding to 16248. We pulverized air-dried leaves (1.5 kg) and macerated them at room temperature with petroleum ether (8L x 2). We filtered, collected, and concentrated the macerated extract using a rotary evaporator (Büchi Labortechnik AG, Flawil, Switzerland) under vacuum at 40 °C to obtain 37.89 g of dried extract (PEAm, 2.52% yield). After we performed the activity-guided isolation, first we examined the antilymphoma activity of PEAm in the induced lymphoma model, after knowing the results, we purified an amount of 975 mg by preparative TLC (silica gel 60F-254 Merck, hexane-EtOAc, 85:15) to obtain thirteen compounds as follows: P1 (22.6 mg); P2 (21.8 mg); P3 (42.6 mg); P4 (41 mg); P5 (29.2 mg); P6 (34.7 mg); P7 (36.4 mg); P8 (56.4 mg); P9 (47.5 mg); P10 (31.2 mg); P11 (26 mg); P12 (53.6 mg); and P13 (168 mg). Once the thirteen compounds had been isolated and purified, we continued with the evaluation of the thirteen compounds in the brine shrimp acute lethality test. We found a higher lethality for compounds P6, P7 and P10, so these were specifically identified. We carried out the extraction and isolation procedure according to the protocol described previously [2].

### 4.3. Identification of Geranylgeraniol, Phytol and Farnesyl Acetate

We performed GC–MS analysis of PEAm using the Agilent GC-MDS (Agilent 220 Technologies, Wilmington, DE, USA) and MassHunter Workstation software, version B.07.05. We used a 5977B gas chromatograph and a 7890B quadrupole mass selective detector. The column used was a capillary column of fused silica and 5% phenyl methyl siloxane HP-5MS (30 m × 0.25 mm internal diameter × 0.25 µm film thickness). The carrier gas was helium at a constant flow of 1.0 mL. We injected a 1.0 μL sample into a 1/10 split injector at 250 °C. The ion source temperature was 230 °C, the quadrupole temperature was 150 °C, and the transfer line was 250 °C. We programmed the mass detector for 70 eV electron impact ionization. We maintained the oven temperature at 130 °C for 2 min initially. We then raised the temperature at a rate of 5 °C/min up to a level of 150 °C, which we maintained for a period of 10 min. We then raised the temperature again, at a rate of 10 °C/min, up to a level of 285 °C, which we again maintained for 10 min. We identified and authenticated the compounds farnesyl acetate (P10), phytol (P7), and geranylgeraniol (P6) using their mass spectra compared to NIST mass spectral libraries [54]. Once we knew the results of the CG-MS analysis, we performed a comparison of the retention factors with the authentic samples of sigma by means of thin layer chromatography of the selected compounds and revealed that with H_2_SO_4_ at 10% (data not shown here), the retention factors for each compound were similar to those of the authentic sample (Sigma-Aldrich). To confirm the identification, we characterized the compounds by spectroscopic (^1^H, and ^13^C NMR) methods (Table 6).

### 4.4. Brine Shrimp Lethality Test

To obtain the LD_50_ of the thirteen compounds isolated by TLC, we performed the brine shrimp assay according to the procedure described by Meyer et al. [31]. We exposed groups of 10 shrimp in brine to concentrations of 0.5, 1, 5, 10 µg mL^−1^ of each product (P1-P13). The concentrations were reached by dissolving the compounds in 2 mL of ethanol, subsequently transferring to different flasks. After evaporating the solvent at room temperature, we introduced the BS and adjusted the volume with artificial seawater to 5 mL. We performed this test in triplicate for each compound. Finally, we observed mortality at 24 h and obtained the values of median lethal concentration (LD_50_) LD_50_ < 100 μg mL^−1^ was considered toxic, LD_50_ > 10 μg mL^−1^ was considered extremely toxic, while LD_50_ < 1000 μg mL^−1^ was considered non-toxic.

### 4.5. Cell-Based Assay

#### 4.5.1. Culture

The cell line U-937 was acquired (ATCC: CRL 1593.2, Middlesex, UK) and characterized as a type of diffuse monocytic NHL; for its propagation two million cells were used and seeded in RPMI 1640 medium (Roswell Park Memorial Institute) supplemented with 2 mM L-glutamine, 10% v/v fetal bovine serum (Thermo), 100 mM 1% sodium pyruvate, and 1% penicillin/streptomycin. The cells were kept in an atmosphere with a concentration of carbon dioxide (CO_2_) at 5% at 37 °C.

#### 4.5.2. Citotoxic Activity

We evaluated the in vitro cytotoxic activity using the cell proliferation reagent WST-1, based on the cleavage of tetrazolium salts to formazan by the enzyme mitochondrial dehydrogenase [55]. For cell proliferation, the Quick Cell Proliferation Kit II (Abcam, Cambridge, UK, Cat. No. ab65475) was used, 96-well plates were used and 2 × 10^5^ cells per well were seeded in a final volume of 100 µL/well of culture medium. After 24 h of incubation, they were treated with terpenoids at different concentrations (60–120 µg mL^−1^), PEAm (60–300 µg mL^−1^), cells with DMSO were considered as control and those treated with MTX as positive control (75–175 µg mL^−1^). After 24 h, viability was assessed by the addition of 10 µL of WST-1 reagent and the cells were further incubated for 2 h at 37 °C and 5% CO_2_. Finally, the absorbance at 440 nm was measured using a microplate reader.

### 4.6. Animals

Male and female mice of the Balb/c strain were provided by the IMSS. The approval of the experimentation procedures was evaluated by the Bioethics Committee of the Specialty Hospital of the National Medical Center “Siglo XXI”, the registration numbers corresponded to R-2020-3601-186 and R-2019-3601-004, they were carried out under the guidelines of the Mexican official standard NOM 0062-ZOO-1999 entitled as Technical Specifications for the Production, Care, and Use of Laboratory Animals [56]. The animals were kept under controlled conditions and periods of light-darkness with a cycle of 12 h at 22 °C ± 2 °C, feeding and water availability was ad libitum.

#### Antilymphoma Activity

We intraperitoneally injected male and female Balb/c mice (20 ± 5 g) with 1 × 10^6^ U-937 cells. We induced lymphoma following the method of Calzada et al. [29]. We randomly formed fifteen groups of male mice and of female mice (n = 6). We carried out a comparative analysis with healthy control (HC, tween 80, 2% *v*/*v* in water), negative control at 28 days (C1), negative control at 65 days (C2) and positive control treated with MTX (0.1, 1.5, and 10 mg kg^−1^) groups to evaluate the extract of petroleum ether from the leaves of *A. macroprophyllata* (100, 200, and 300 mg kg^−1^) and the terpenoids FA, PT, and Gg (1.5, 10, and 15 mg kg^−1^). After cell inoculation, we observed all animals for the following 28 days, after which we orally administered treatments for 9 days. Over the following 28 days, we recorded the weight and survival of the animals. Finally, at day 65, we sacrificed the animals and removed and weighed the left and right axillary and inguinal lymph nodes. We determined antilymphoma activity by comparing the total weight of lymph nodes in each group against that of C2.

### 4.7. Acute Oral Toxicity

We performed the acute oral toxicity test according to the guidelines described by the Organization for Economic Cooperation and Development (OCDE) guideline 423 for the evaluation of the acute oral toxicity of chemical substances [40]. Twenty-four female mice of the Balb/c strain were randomly used to form 8 groups (n = 3). Before administering the compounds, the animals were fasted for 12 h and had free access to water. We administered a single dose of FA, PT, and MTX orally. The control group was administered with only the vehicle (Tween 80 at 2% in water) and for the compounds, doses of 500 and 1000 mg kg^−1^ were administered. The animals were observed for the next 4 h, registering any sign of toxicity (seizures, tremors, diarrhea, sleep, etc.) and/or death, up to 14 days. Subsequently, we sacrificed the animals and extracted the organs (stomach, intestines, spleen, liver, and kidneys) were extracted, reporting the pathological changes, if any. The results were compared with the control group. With the data obtained by this test, we determined the median lethal dose (LD_50_) for each compound, taking into account the classification of acute systemic toxicity recommended by the OECD (40) following categories: category 1, very toxic ≤ 5 mg/kg; category 2, toxic > 5 and ≤ 50 mg/kg; category 3, harmful > 50 and ≤ 300 mg/kg; category 4, low risk > 300 and ≤2000 mg/kg; and category 5, safe or without label > 2000 mg/kg.

### 4.8. Molecular Docking Studies of Geranylgeraniol, Farnesyl Acetate, and Phytol

The chemical structure of ligands farnesyl acetate (CID: 638500), phytol (CID: 5280435) geranylgeraniol (CID: 5281365) and Hydroxymethylglutaril-CoA-substrate (CID: 445127) were retrieved from the chemical library PubChem (https://pubchem.ncbi.nlm.nih.gov/) (accessed on 13 September 2022), these were optimized and submitted to energetic and geometrical minimization using Avogadro software [57]. Hydroxymethylglutaril-CoA reductase (RCSB, PDB ID: 1DQ9) was used as a target of the study. This was retrieved from the Protein Data Bank (http://www.rcsb.org/) (accessed on 13 September 2022), total molecules of water and ions no needed to catalytic activity were stripped to preserve the entire protein. All polar hydrogen atoms were added, ionized in a basic environment (pH = 7.4), and Gasteiger charges were assigned, the computed output topologies from the previous steps were used as input files to docking simulations.

The molecular docking experiments were carried out using Autodock 4.2 software [58], the search parameters were as follows: a grid-base procedure was employed to generate the affinity maps delimiting a grid box of 126 × 126 × 126 Å3 in each space coordinate, with a grid points spacing of 0.375 Å, the Lamarckian genetic algorithm was employed as scoring function with randomized initial population of 100 individuals and maximum number of energy evaluations of 1 × 107 cycles, the analysis of the interactions in the enzyme/inhibitor complex was visualized with PyMOL software (The PyMOL Molecular Graphics System, Ver 2.0, Schrödinger, LLC, DeLano Scientific, San Carlos, CA, USA). The validation of the molecular docking was carried out by re-docking the co-crystallized ligand in the receptor (HMG-CoA). The lowest energy pose of the co-crystallized ligand was superimposed and it was observed whether it maintained the same bind position. The RMSD was calculated and a reliable range within of 2 Å is reported.

### 4.9. Statistical Analysis

The results were expressed as mean ± standard error of six measurements. The results were analyzed using the GraphPad Prisma version 5 program (GraphPad software, San Diego, CA, USA), performing one-way ANOVA, as well as multiple comparison tests using Dunnet with a value of *p* ˂ 0.05 to establish that there were significant differences between the study groups. The CC_50_, LD_50_, and LC_50_ were calculated by linear interpolation of the percentage mortality values for each concentration.

## 5. Conclusions

A complete analysis of our results suggests that the petroleum ether extract obtained from *A. macroprophyllata* and its acyclic terpenoids farnesyl acetate, phytol and geranylgeraniol may have potential for the future treatment for cancer. These molecules could be candidates as safe anticancer agents. In addition, their mechanism of action could be mediated by affecting the activity of HMG-CoA reductase. However, more studies are needed to confirm the mechanism by which such terpenoids achieve their antilymphoma effects.

## Figures and Tables

**Figure 1 molecules-27-07123-f001:**
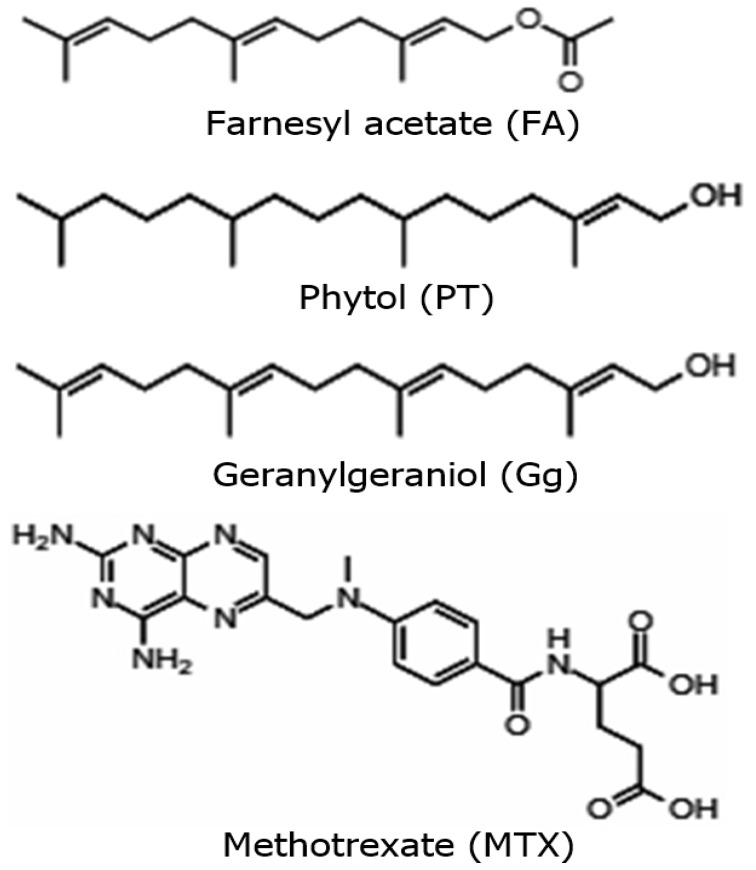
Structures of geranylgeraniol, phytol, and farnesyl acetate obtained of petroleum ether extract from *A. macroprophyllata* leaves (PEAm) and methotrexate.

**Figure 2 molecules-27-07123-f002:**
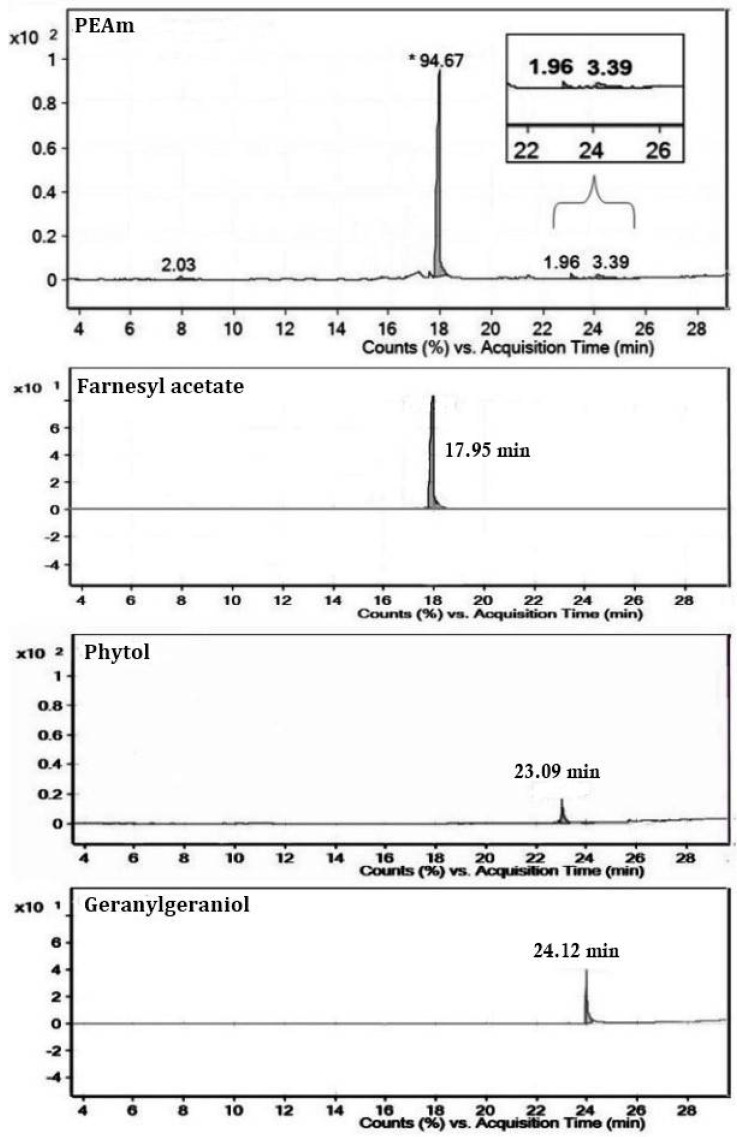
Gas chromatography-mass spectrometry analysis of petroleum ether extract from *A. macroprophyllata* leaves (PEAm) and farnesyl acetate, phytol and geranylgeraniol standards. The *x*-axis indicates the retention time in minutes, while the *y*-axis indicates the peak % signal intensity. * *p* < 0.05.

**Figure 3 molecules-27-07123-f003:**
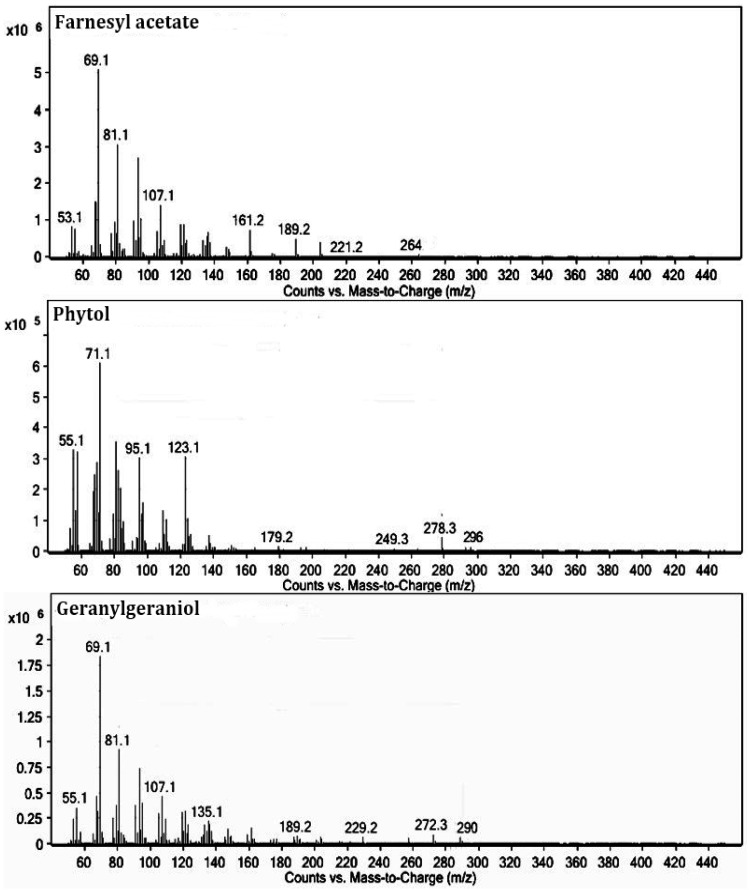
Mass spectrum of farnesyl acetate (P10), phytol (P7), and geranylgeraniol (P6).

**Figure 4 molecules-27-07123-f004:**
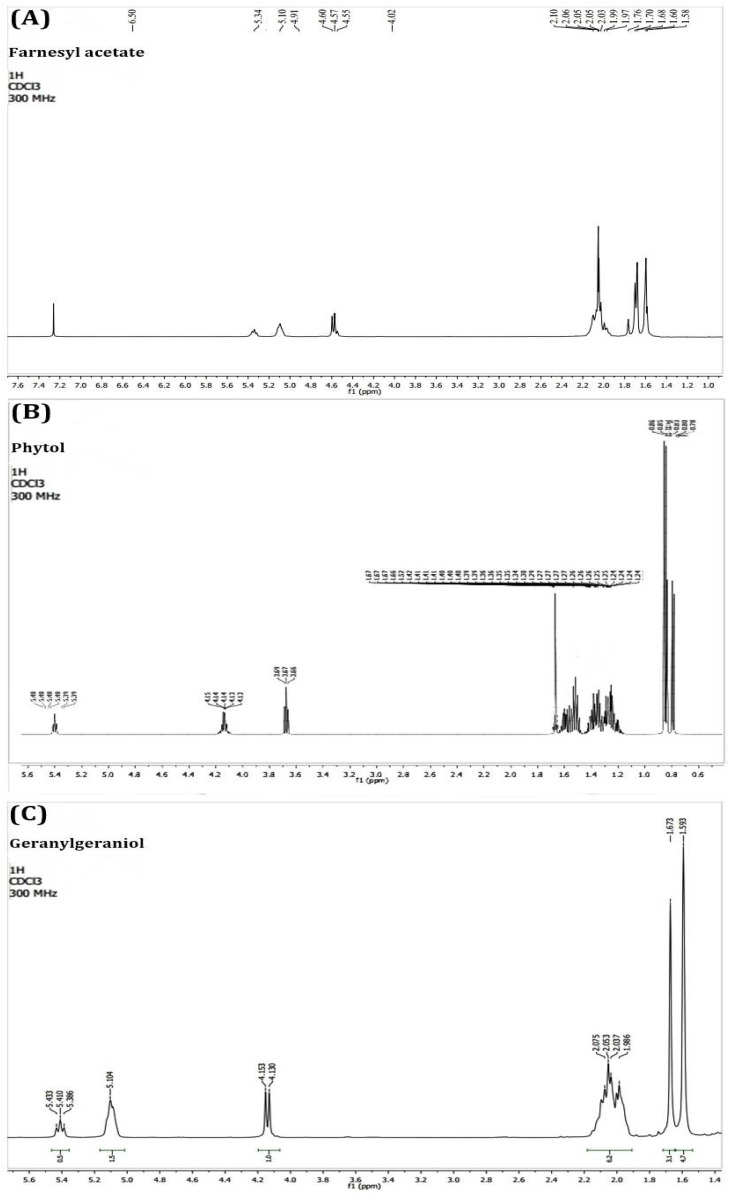
^1^H-NMR spectra of acyclic terpenoids farnesyl acetate (**A**), phytol (**B**) and geranylgeraniol (**C**).

**Figure 5 molecules-27-07123-f005:**
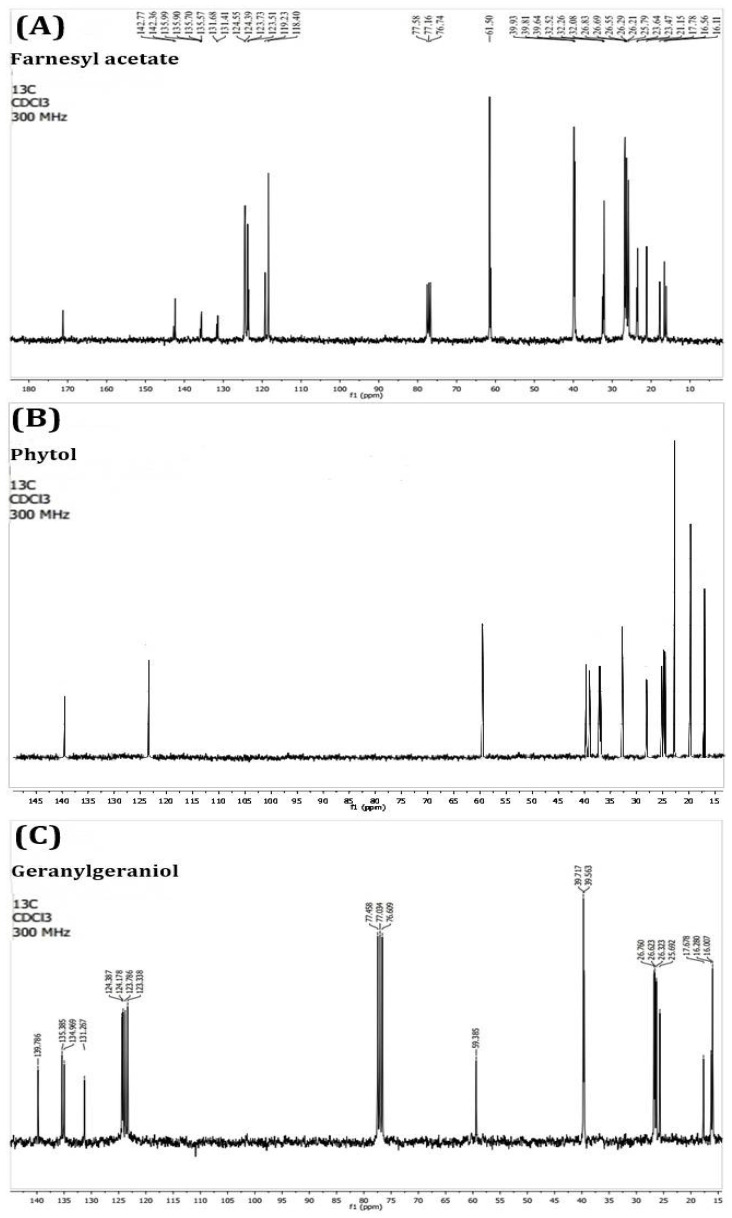
^13^C-NMR spectra of acyclic terpenoids farnesyl acetate (**A**), phytol (**B**) and geranylgeraniol (**C**).

**Figure 6 molecules-27-07123-f006:**
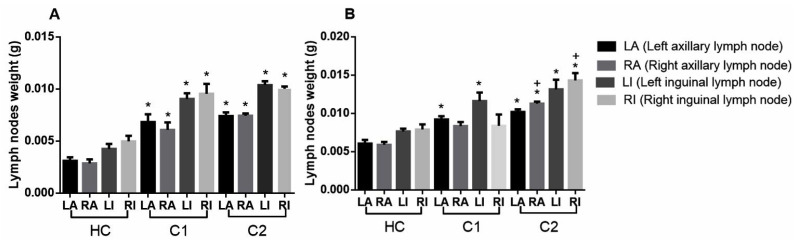
Weights (in g) of left axillary (LA), right axillary (RA), left inguinal (LI) and right inguinal (RI) lymph nodes of female (**A**) and male (**B**) mice, compared with healthy control (HC), control without treatment at 28 days (C1) and control without treatment at 65 days (C2) groups. Results obtained by ANOVA one-way analysis followed by Dunnett’s test for multiple comparison. Data are expressed as mean ± SEM, (n = 6); * *p* < 0.05 vs. HC, + *p* < 0.05 vs. C1.

**Figure 7 molecules-27-07123-f007:**
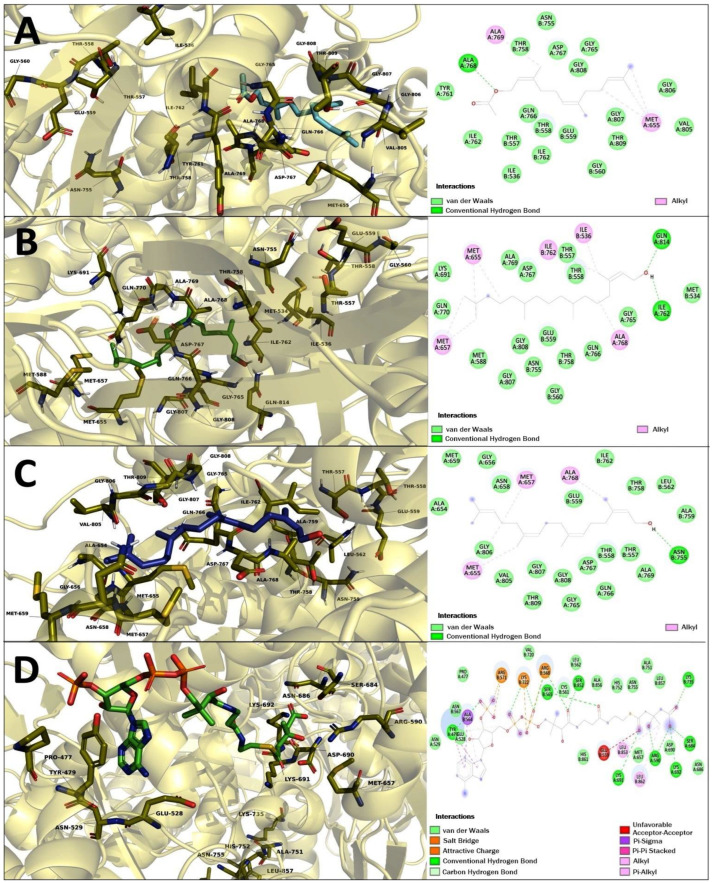
Molecular Docking of the terpenoids bound to HMG-CoA reductase, (**A**) farnesyl acetate, (**B**) phytol, (**C**) geranylgeraniol and (**D**) HMG-CoA substrate.

**Table 1 molecules-27-07123-t001:** LD_50_ of the thirteen compounds isolated (P1-P13) by preparative thin-layer chromatography from petroleum ether extract of *A. macroprophyllata* leaves and methotrexate (MTX) obtained from the BSL test.

Sample	LD_50_ (µg mL^−1^) ^a^
PEAm	18.84 ± 0.58 *
P1	20.75 ± 0.30
P2	8.07 ± 0.59 *
P3	9.06 ± 0.45 *
P4	45.32 ± 0.33 *
P5	8.58 ± 0.28 *
Gg (P6)	1.41 ± 0.42 *
PT (P7)	3.03 ± 0.33 *
P8	25.88 ± 0.31
P9	9.90 ± 0.41 *
FA (P10)	5.82 ± 0.58 *
P11	13.34 ± 0.34 *
P12	13.79 ± 0.56 *
P13	9.98 ± 0.88 *
MTX	24.66 ± 0.27

^a^ LD_50_: Median lethal dose (causing 50% of *A. salina* death), calculated by linear regression analysis of percentage mortality against concentration. Data are expressed as mean ± SEM, (n = 3). Geranylgeraniol (P6), phytol (P7) and farnesyl acetate (P10) showed more lethality compared to methotrexate (MTX). * *p* < 0.05 vs. MTX.

**Table 2 molecules-27-07123-t002:** GC-MS, retention times, and percent in area of geranylgeraniol (Gg), phytol (PT), and farnesyl acetate (FA) identified from petroleum ether extract from *A. macroprophyllata* leaves.

Compound Name	IUPAC Name	R.T * (Min)	Area %	Molecular Weight	Molecular Formula
Farnesyl acetate	[(E,E)-3,7,11-trimethyldodeca-2,6,10-trienyl] acetate	17.95	94.67	264	C_17_H_28_O_2_
Phytol	(2E,7R,11R)-3,7,11,15-tetramethylhexadec-2-en-1-ol	23.09	1.96	296	C_20_H_40_O
Geranylgeraniol	(2E,6E,10E)-3,7,11,15-tetramethylhexadeca-2,6,10,14-tetraen-1-ol	24.12	3.39	290	C_20_H_34_O

* R.T: Retention time.

**Table 3 molecules-27-07123-t003:** ^1^H and ^13^C (300 MHz) NMR data of compounds geranylgeraniol (P6), phytol (P7) and farnesyl acetate (P10) in CDCl3.

Position	Geranylgeraniol (P6)		Phytol (P7)		Farnesyl Acetate (P10)	
	δ_H_, Mult. (*J* in Hz)	δ_C_, Type	δ_H_, Mult. (*J* in Hz)	δ_C_, Type	δ_H_, Mult. (*J* in Hz)	δ_C_, Type
**1**	4.14 (2H, m, 6.9)	59.4 (CH_2_)	4.14 (2H, m, 0.2)	59.4 (CH_2_)		171.21 (C)
**2**	5.41 (1H, t. 7.29)	123.3 (CH)	5.4 (1H, t. 0.1)	123.4 (CH)	2.06 (3H, s)	21.15 (CH_3_)
**3**		139.8 (C)		25.1 (C)	4.57 (2H, m, 8.1)	61.5 (CH_2_)
**4**	2.148 (2H, m, 6.6)	39.7 (CH_2_)	1.59 (2H, m, 0.01)	39.7 (CH_2_)	5.34 (1H, m, 6.1)	118.4 (CH_2_)
**5**	2.148 (2H, m, 6.6)	39.6 (CH_2_)	1.52 (2H, s)	25.2 (CH_2_)		142.3 (C)
**6**	5.10 (1H, m, 0.6)	123.7 (CH)	5.10 (2H, m, 0.6)	36.8 (CH_2_)	2.10 (2H, m, 3.1)	39.93 (CH_2_)
**7**		135.4 (C)	1.57 (1H, d, 0.6)	32.71 (CH)	2.10 (2H, m, 3.1)	26.6 (CH_2_)
**8**	2.148 (2H, m, 6.6)	26.7 (CH_2_)	1.24 (2H, m, 0.06)	37.2 (CH_2_)	4.91 (1H, d, 2.8)	123.7 (CH_2_)
**9**	2.148 (2H, m, 6.6)	26.6 (CH_2_)	3.6 (2H, m, 0.04)	24.5 (CH_2_)		135.5 (C)
**10**	5.10 (1H, m, 0.6)	124.1 (CH)	1.24 (2H, m, 0.06)	37.0 (CH_2_)	1.98 (2H, t, 0.04)	39.8 (CH_2_)
**11**		134.9 (C)	1.52 (1H, d, 0.7)	32.72 (CH)	2.06 (2H, m, 4.2)	26.21 (CH_2_)
**12**	2.148 (2H, m, 6.6)	26.3 (CH_2_)	1.32 (2H, s)	36.9 (CH_2_)	5.10 (1H, t, 2.2)	124.3 (CH_2_)
**13**	2.148 (2H, m, 6.6)	25.7 (CH_2_)	1.29 (2H, m, 0.6)	24.7 (CH_2_)		131.5 (C)
**14**	5.10 (1H, m, 0.6)	124.4 (CH)	1.28 (2H, s)	38.9 (CH_2_)	1.64 (3H, m, 7)	25.79 (CH_3_)
**15**		131.3 (C)	1.52 (1H, d, 0.02)	28.0 (CH)	1.62 (3H, m, 2.1)	21.15 (CH_3_)
**16**	1.673 (3H, s)	17.67 (CH_3_)	0.79 (3H, d, 0.1)	22.72 (CH_3_)	1.62 (3H, m, 2.1)	16.1 (CH_3_)
**17**	1.673 (3H, s)	16.28 (CH_3_)	0.84 (3H, 0.1)	22.7 (CH_3_)	1.68 (3H, d, 3.1)	17.78 (CH_3_)
**18**	1.593 (3H, s)	16 (CH_3_)	0.85 (3H, s)	19.7 (CH_3_)		
**19**	1.593 (3H, s)	15.87 (CH_3_)	0.86 (3H, s)	19.6 (CH_3_)		
**20**	1.593 (3H, s)	15.97 (CH_3_)	1.67 (3H, d, 0.6)	17 (CH_3_)		

**Table 4 molecules-27-07123-t004:** Median cytotoxic concentration calculated after 24 h of exposure against U-937 cells from petroleum ether extract and its terpenoids geranylgeraniol (Gg), phytol (PT), and farnesyl acetate (FA) obtained and identified from *A. macroprophyllata* leaves.

Sample	CC_50_ (µg mL^−1^) ^a^
PEAm	298.30 ± 2.87
	CC50 (µM) ^a^
FA	0.275 ± 0.08
PT	0.296 ± 0.07
Gg	0.395 ± 0.005 *
MTX	0.243 ± 0.007

^a^ CC_50_: Median cytotoxic concentration causing 50% cell death. Calculated by linear regression analysis of percentage mortality against concentration. Data are expressed as mean ± SEM, (n = 3). * *p* < 0.05 vs. MTX.

**Table 5 molecules-27-07123-t005:** Results of the antilymphoma activity after administration of petroleum ether extract, geranylgeraniol (Gg), phytol (PT), and farnesyl acetate (FA) obtained from the leaves of *A. macroprophyllata* on male and female mice inoculated with U-937 cells.

Treatment	ED_50_ (mg kg^−1^) ^a^	
	Female	Male
PEAm	180.42 ± 1.65 *	166.41 ± 3.8 *
FA	5.89 ± 0.39 *	5.09 ± 0.66 *
PT	6.71 ± 0.31 *	5.83 ± 0.50 *
Gg	7.22 ± 0.51 *	6.98 ± 0.57 *
MTX	1.31 ± 0.34	0.99 ± 0.024

^a^ ED_50_: Median effective dose (causing 50% of population with the desired pharmacological effect). Linear regression analysis of percentage of inhibition of lymph node growth against the dose administered for each treatment. The results are presented as mean ± SEM, (n = 6). * *p* < 0.05 vs. MTX.

**Table 6 molecules-27-07123-t006:** Acute oral toxicity from petroleum ether extract and its terpenoids geranylgeraniol (Gg), phytol (PT), and farnesyl acetate (FA) obtained of the leaves from *A. macroprophyllata* leaves.

Sample	LD_50_ (mg/kg) ^a^	TI ^b^ Female	TI ^b^ Male
PEAd	>3000	16.62	18.02
FA	>1000	169.77	196.46
PT	>1000	149.03	171.52
Gg	742.17 ± 0.34	138.5	143.26
MTX	335.04 ± 0.39	255.75	338.42

^a^ LD_50_: Median lethal dose (causing 50% of animal death). Calculated by linear regression analysis of percentage mortality against the dose administered for each treatment. The results are presented as mean ± SEM, (n = 3). ^b^ TI: therapeutic index calculated as LD_50_ (acute oral toxicity)/ED_50_ (antilymphoma activity).

**Table 7 molecules-27-07123-t007:** Interactions of farnesyl acetate (FA), Phytol (PT), geranylgeraniol (Gg) and HMG CoA with residues on the binding sites of HMG-CoA reductase.

Compound	HMG-CoA Reductase			
	ΔG (kcal-mol^−^^1^)	H-BR	NPI	RMSD
Farnesyl acetate	−7.38	Ile 536, Thr 557, Thr 558, Glu 559, Gly 560, Asn 755, Thr 758, Tyr 761, Ile 762, Gly 765, Gln 766, Asp 767, Ala 768, Val 805, Gly 806, Gly 807, Gly 808, Thr 809	Met 655, Ala 769	-
Phytol	−7.55	Met 534, Thr 557, Thr 558, Glu 559, Gly 560, Met 588, Lys 691, Asn 755, Thr 758, Ile 762, Gly 765 Gln 766, Asp 767, Ala 769, Gln 770, Gly 807, Gly 808, Gln 814	Ile 536, Met 655, Met 657, Ile 762, Ala 768	-
Geranylgeraniol	−7.95	Glu 559, Leu 562, Thr 557, Thr 558, Ala 654, Gly 656, Asn 658, Met 659, Asn 755, Thr 758, Ala 759, Ile 762, Gly 765, Gln 766, Asp 767, Ala 769, Val 805, Gly806, Gly 807, Gly 808, Thr 809	Met 655, Met 657, Ala 768	-
HMG-CoA- substrate	−9.21	Pro 477, Tyr 479, Glu 528, Asn 529, Cys 561, Leu 562, Ser 565, Asn 567, Arg 590, Met 657, Ser 684, Asn 686, Asp 690, Lys 691, Lys 692, Val 720, Lys 735, Ala 751, His 752, Asn 755, Ser 852, Ala 856, Leu 857, His 861	Tyr 479, Ala 564, Arg 568, Arg 571, Lys 722, Leu 853, Leu 862	1.10

ΔG: Binding energy (kcal/mol^−1^); H-BR: H-binding residues; NPI: Nonpolar interactions; Asp: Aspartate; Asn: Asparagine; Arg: Arginine; Gln: Glutamine; Lys: Lysine; Thr: Threonine; Ser: Serine; Trp: Tryptophan; Leu: Leucine; His: Histidine; Gly: Glycine; Glu: Glutamic acid; Ile: Isoleucine; Tyr: Tyrosine; Phe: Phenylalanine.

## Data Availability

The data presented or additional data in this study are available on request from the corresponding author.

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
