# Peer review of "Understanding the Antilymphoma Activity of Annona macroprophyllata Donn and Its Acyclic Terpenoids: In Vivo, In Vitro, and In Silico Studies"

_molecules, 2022, doi:10.3390/molecules27207123_

Round 1
Reviewer 1 Report
In the manuscript molecules-1910823, the authors isolated several terpenoids, namely farnesyl acetate, phytol, and geranylgeraniol from A. diversifolia leaves and investigated cytotoxicity and anti-lymphoma activity. The topic and novelty of this manuscript seem to be interesting and meet the journal’s scope. Nevertheless, I’m sorry to tell you that it is hard to consider its acceptance for publication in the current level, and more experiments and results are needed to meet the academic and scientific levels of MDPI Molecules. As a temporary reviewer, I’d like to suggest the authors some flaws of this manuscript to be reinforced and hope the authors significantly revised their manuscript for improving the quality of this manuscript as indicated below.
1. The authors should provide logical reasons why they have chosen their methodologies. Why they chose GC-MS rather than other analysis, such as LC-MS, to identify the secondary metabolites in their extract? GC-MS is an analytical method that is used to detect volatile compounds, but LC-MS can apply to determine a wide range of phytochemical. What is the specific reason that they chose U-937 cells, rather than other Hodgkin lymphoma cell lines (L428, L540, L591, L1236, KM-H2, HDLM-2, UHO-1, SUP-HD1, and DEV) for evaluating cytotoxic effect of their sample? If possible, please provide relevant references.
2. The authors are needed to mention why they focused on FA, PT, and Gg rather than other compounds, such as polyphenol, flavonoids, and other phytochemicals. What is the special reason to be studied? Also, the quantitative results of aforementioned compounds should be determined in PEAd and provided in the main text.
3. The authors provided TIC obtained from the extract and standard references to identify the active compounds present in their extract. But, providing only TIC without MS and MS/MS does not prove evidence of identification of the active compounds. Further, I could not believe the R.T. of farnesyl acetate (16.96 min) provided by authors. Please reconfirm the Figure 1.
4. The English uses should be improved significantly. There are a lot of spelling and grammatical errors in the main text and figure artworks. Ex) L63, made up pf? L101, Citotoxicity assay?, L313, Quimicos? Chemicals?
5. The resolution and quality of most figures should be notably improved.
6. All figure and table legends should be rewritten for clearly understand of the authors findings without reading description in the main text. Good legends can allow the readers to understand the authors results easily.
7. The authors mentioned that the PEAd transpired a total of 16 chemical constituents in Figure 1, but how can we identify them?
8. The units provided need to be unified throughout the main text. mg/kg? mg/kg-1?
9. The abbreviations used should be defined when it appeared first in the main text excluding Abstract section. Do not use abbreviation when the terms were appeared just one time throughout the manuscript.
10. Statistical analysis should be performed using appropriate multiple comparison methods in all figures and tables.
11. The title needs to be modified because it is too comprehensive to be supported by the results.
Author Response
Dear reviewer, in the attached file the answers to your suggestions.
Thank you for your suggestions and comments that enrich our work allowing us to make the necessary corrections.
Next, I send you the corrections for each of your observations.
- The authors should provide logical reasons why they have chosen their methodologies. Why they chose GC-MS rather than other analysis, such as LC-MS, to identify the secondary metabolites in their extract? GC-MS is an analytical method that is used to detect volatile compounds, but LC-MS can apply to determine a wide range of phytochemical. What is the specific reason that they chose U-937 cells, rather than other Hodgkin lymphoma cell lines (L428, L540, L591, L1236, KM-H2, HDLM-2, UHO-1, SUP-HD1, and DEV) for evaluating cytotoxic effect of their sample? If possible, please provide relevant references.
Response. In the line 283-288, we report the following. We applied a presumptive identification thin-layer chromatography method with 10% H2SO4 and found that all products exhibited a brown-purple stain characteristic of terpene molecules (data not presented here). We characterized these by GC-MS analysis (Table 6, Figure 4), taking particular account of the volatile compounds of a terpenic nature, due to previous studies had reported the presence of acyclic terpenoids in this specie [29] and also terpenoids with antitumor activity [2, 30].
Response. In the line 255-258, we report the following. We used this cell line because it is more aggressive, has a histiocytic subtype, and exhibits diffuse proliferation; moreover, it is routinely used for the evaluation of antitumor and cytotoxic activity in natural products [21,22,23,24].
- The authors are needed to mention why they focused on FA, PT, and Gg rather than other compounds, such as polyphenol, flavonoids, and other phytochemicals. What is the special reason to be studied? Also, the quantitative results of aforementioned compounds should be determined in PEAd and provided in the main text.
Response. In the line 275-283, we report the following. We obtained thirteen products which we then subjected to lethality evaluation by means of an BSL assay which has been previously used as a bioindicator in the screening of possible molecules with antitumor potential [2, 27, 28]. Our results showed that Gg (P6), PT (P7) and FA (P10) induced a higher dose-dependent lethality with LD50 values of <10 µg mL−1, representing a lethality greater than that of MTX (LD50 = 24.66±0.27 µg mL−1) (Table 1). In line with the findings of previous studies, our results suggested that these molecules might have antitumor potential [28, 29, 30], so we carried out a further investigation of P6, P7 and P10.
- The authors provided TIC obtained from the extract and standard references to identify the active compounds present in their extract. But, providing only TIC without MS and MS/MS does not prove evidence of identification of the active compounds. Further, I could not believe the R.T. of farnesyl acetate (16.96 min) provided by authors. Please reconfirm the Figure 1.
Response. In the line 285 we reported that P6, P7 and P10 were characterized by GC-MS analysis (Table 6, Figure 4) and in line 288 we reported the spectroscopic (1H, and 13C NMR) data (Table 7). In the figure 4 and Table 6 we report that FA was identified with a RT of 17.95 min.
- The English uses should be improved significantly. There are a lot of spelling and grammatical errors in the main text and figure artworks. Ex) L63, made up pf? L101, Citotoxicity assay?, L313, Quimicos? Chemicals?
Response. All the manuscript was submitted to English editing and all the grammatical errors was corrected.
- The resolution and quality of most figures should be notably improved.
Response. The quality of all images has been improved
- All figure and table legends should be rewritten for clearly understand of the authors findings without reading description in the main text. Good legends can allow the readers to understand the authors results easily.
Response. Figure and table legends were rewritten.
- The authors mentioned that the PEAd transpired a total of 16 chemical constituents in Figure 1, but how can we identify them?
Response. The extract transpired 16 constituents, however, in this work we only focus on acyclic terpenoids since the other compounds will be included in other works. Therefore, this paragraph was modified by mentioning only the identification of the compounds evaluated.
- The units provided need to be unified throughout the main text. mg/kg? mg/kg-1?
Response. The units were unified as mg kg-1
- The abbreviations used should be defined when it appeared first in the main text excluding Abstract section. Do not use abbreviation when the terms were appeared just one time throughout the manuscript.
Response. The abbreviations were first defined in the main text.
- Statistical analysis should be performed using appropriate multiple comparison methods in all figures and tables.
Response. Multiple comparisons of all figures and tables were performed.
- The title needs to be modified because it is too comprehensive to be supported by the results.
Response. Title changed to “Understanding the Antilymphoma Activity of Annona macroprophyllata Donn and Its Acyclic Terpenoids: In vivo, In vitro, and In silico Studies”.
Additionally, the manuscript was sent to English editing in MDPI English editing services

Reviewer 2 Report
The antilymphoma activity of single natural compounds and plant extract was done in the current manuscript titled " Understanding the Antilymphoma Activity of Annona diversifolia Safford: In vivo, In vitro, Phytochemical, and Toxicological Studies." The research work is particularly impressive despice the topic is not novel, and the authors conducted many studies in the same topic using different plant species belonging to Annona genera with the same experimental design. The manuscript must be revised, along with proper formatting and the correction of numerous grammar (was/were/is), spelling, and scientific terminology errors.
General comments:
1. The abstract must be rechecked add important results must be described (with values).
2. Many references must be added by the end of Line 45, Line 71.
3. Authors have to increase the resolution of all figures
4. Authors have to add statistical analysis for Figure 3, Figure 4; Tables (2, 3, 4, 5)
5. Authors should discuss the obtained results with Annona diversifolia with previous results from others plant species belonging to Annona genus (Published by the same team)
6. In silico approach have to done to confirm the interactions between the identified phytoconstituents with specific target proteins as previously done in previous works.
7. Authors have to include the complete list of the identified molecule in the tested extract
8. What about the pic at 2.03 min?, and pics between 16 and 18 min
9. What about the yield of extraction (Material and methods section)?
10. Add more details about the extraction and isolation procedure
11. A list of abbreviations must be added
Specific comments are as follows:
· Some spelling errors (line 63 (is made up of), 64 (than fifty….) page 2/14
· Farnesyl acetate R.T. was 16.96 min (See table 1): please verify with figure 1.
· Page 8/14: Line 35: plant species
· Page 9/14: Line 64: Please delete the comma or the dot?
· Annona have to be written A. after the first citation in the manuscript: please revise through the manuscript
Author Response
Thank you for your suggestions and comments that enrich our work allowing us to make the necessary corrections.
Next, I send you the corrections for each of your observations.
- The abstract must be rechecked add important results must be described (with values).
Response. In line 21-45 the abstract was rewritten, and the most important results were described.
- Many references must be added by the end of Line 45, Line 71.
Response. In the line 57 and 84 the corresponding references were added.
- Authors have to increase the resolution of all figures.
Response. The quality of the images has been improved
- Authors have to add statistical analysis for Figure 3, Figure 4; Tables (2, 3, 4, 5).
The corresponding statistical analysis was added to figure 1 and 2.
- Authors should discuss the obtained results with Annona diversifolia with previous results from others plant species belonging to Annona genus (Published by the same team)
Response. In line 267-269, the results were discussed with the previously published work of the species Annona muricata.
- In silico approach have to done to confirm the interactions between the identified phytoconstituents with specific target proteins as previously done in previous works.
Response. In silico studies has been made to the compounds isolated, in line 197-204, we reported that we carried out a molecular docking using the HMG-CoA reductase enzyme as a target, in the knowledge that this is the regulatory enzyme of the mevalonate pathway. This pathway contributes to the modulation of various pathways involved in cancer progression through the prenylation of RAS, RAC, Rho and GTPases proteins related to cell growth and cell proliferation in cancer [46,47]. In addition, terpenoids can bring about an impact on the enzyme HMG-CoA reductase [48, 40, 14].
- Authors have to include the complete list of the identified molecule in the tested extract.
Response. In this work only the identification of acyclic terpenoids was considered.
- What about the pic at 2.03 min?, and pics between 16 and 18 min.
Response. At 2.03 min a bicyclic sesquiterpene was identified and at 16-18 min FA was identified with a RT of 17.95 min.
- What about the yield of extraction (Material and methods section)?
Response. In line 437-445 the yields of PEAm and of the isolated compounds were added.
- Add more details about the extraction and isolation procedure.
Response. In the line 437-445 added more details on the extraction and isolation procedure.
- A list of abbreviations must be added.
Response. We add the abbreviations in the text and are defined when they first appear. The journal does not include an abbreviations section.
Specific comments are as follows:
- Some spelling errors (line 63 (is made up of), 64 (than fifty….) page 2/14
Response. In the line 75 was corrected and in the line 75 the word fifty was modified
- Farnesyl acetate R.T. was 16.96 min (See table 1): please verify with figure 1.
Response. Farnesyl acetate R.Twas corrected to 17.95 min (See Table 6 and Figure 4)
- Page 8/14: Line 35: plant species
Response. In the line 293-294 the paragraph was modified.
- Page 9/14: Line 64: Please delete the comma or the dot?
Response. In the line 370 the point was deleted.
- Annona have to be written A. after the first citation in the manuscript: please revise through the manuscript
Response. All the manuscript was revised and modified to be homogeneous. After the first citation, A. macroprophyllata was written throughout the manuscript.
Additionally, the manuscript was sent to English editing in MDPI English editing services.

Reviewer 3 Report
After reviewing the manuscript, we revealed the comments bellow:
Line 2 and 3 : After checking in the database world flora, we found that the scientific name of th plant for your experiment “Annona diversifolia” is the synonym of the accepted name “Annona macroprophyllata Donn. Sm”. You have to change the name of your plant with the accepted one
Line 20 (Abstract): It missed the introduction or background
Line 36 : the scientific name should not be considered as a keyword, you have to delete it.
Line 45 and 56-58 : You have to add references related to these informations
Line 99 and 100: You have to add the sources of these chemical compounds
Line 114-116: The quality of the figures is bad; you have to improve the resolution. You have to check this in all manuscript.
Line 135-138: This sentence should be included in the section on Materials and methods. Furthermore, you should be more succinct and explicit.
Line 333-346 and Line 370-379 : You have to add the references related to this protocol for your experiment.
Line 418-420: You have to add the type classification used for this purpose also the reference related to this classification
For your statistical analysis: ANOVA? Were the data found to be normal?
Conclusion: The conclusion should highlight the major findings of the present study and should be more precise about the finding
Author Response
Thank you for your suggestions and comments that enrich our work allowing us to make the necessary corrections.
Next, I send you the corrections for each of your observations.
Line 2 and 3 :After checking in the database world flora, we found that the scientific name of th plant for your experiment “Annona diversifolia” is the synonym of the accepted name “Annona macroprophyllata Donn. Sm”. You have to change the name of your plant with the accepted one.
Response. The name is changed to Annona macroprophyllata.
Line 20 (Abstract): It missed the introduction or background
Response. In the line 20-21 the introduction is added.
Line 36: the scientific name should not be considered as a keyword, you have to delete it.
Response. In the line 49 the scientific name is removed.
Line 45 and 56-58: You have to add references related to these informations
Response. In the line 57 and 84 the references were added.
Line 99 and 100: You have to add the sources of these chemical compounds
Response. In the line 319 the chemical compounds were added.
Line 114-116: The quality of the figures is bad; you have to improve the resolution. You have to check this in all manuscript.
Response. The quality of the images has been improved
Line 135-138: This sentence should be included in the section on Materials and methods. Furthermore, you should be more succinct and explicit.
Response. All the section was modified and added to Materials and Methods section.
Line 333-346 and Line 370-379 : You have to add the references related to this protocol for your experiment.
Response. In the line 463 and line 469 the.references were added.
Line 418-420: You have to add the type classification used for this purpose also the reference related to this classification
Response. In the line 541 the classification and reference used were added.
For your statistical analysis: ANOVA? Were the data found to be normal?
Response. Yes, for the one-way ANOVA statistical analysis the data were normal.
Conclusion: The conclusion should highlight the major findings of the present study and should be more precise about the finding
Response. In the line 570-585 the conclusion was modified, highlighting the main findings.
Additionally, the manuscript was sent to English editing in MDPI English editing services

Round 2
Reviewer 1 Report
I admit and respect the authors effort to improve the quality of their manuscript reflecting my previous suggestions/recommendations. Although some technical improvements have been made by the authors, unfortunately, I think significant flaws remain in the amendment that have not yet been clarified. Above all, I wonder why the authors had to change their sample name. The authors did not address why the sample used has been changed from A. diversifolia Safford in the original manuscript into A. macroprophyllata Donn in the present one. Second, the description of retention time of farnesyl acetate on GC-TIC which was pointed by this reviewer was simply changed without any explanation. Was it just a typo? Unfortunately, these behaviors can be an significantly important issue that makes the authors doubt the legitimacy and integrity of their manuscript. Third, as I previously suggested, I strongly insist that the authors need to provide evidence whether the key compounds are actually existed in the extracts. These concerns can be fully elucidated by providing comparison spectra obtained between extracts and references through MS or NMR. Fourth, quantitative results of farnesyl acetate, phytol, and geranylgeraniol should be provided using GC or NMR analysis, not a lethality test. Unfortunately, I think this work is unsuitable to be published in Molecules.
Author Response
REVIEWER 1
Dear reviewer, we appreciate your suggestions, comments, and observations that helped us improve the manuscript.
Next, I send you the corrections for each of your observations.
Above all, I wonder why the authors had to change their sample name. The authors did not address why the sample used has been changed from A. diversifolia Safford in the original manuscript into A. macroprophyllata Donn in the present one
Response. We changed the name of the sample at the suggestion of a reviewer, "Annona diversifolia Safford" is the synonym of "Annona macroprophyllata Donn" according to the database world flora.
Second, the description of retention time of farnesyl acetate on GC-TIC which was pointed by this reviewer was simply changed without any explanation. Was it just a typo?
Response. We had an error when writing in our manuscript the retention time of farnesyl acetate from the GC-MS analysis.
Third, as I previously suggested, I strongly insist that the authors need to provide evidence whether the key compounds are actually existed in the extracts. These concerns can be fully elucidated by providing comparison spectra obtained between extracts and references through MS or NMR.
Response. We provide the spectra in the following figures.
Figure 3. Mass spectrum of farnesyl acetate (P10), phytol (P7), and geranylgeraniol (P6).
Figure 4. 1H-NMR spectra of acyclic terpenoids farnesyl acetate (A), phytol (B) and geranylgeraniol (C).
Figure 5. 13C-NMR spectra of acyclic terpenoids farnesyl acetate (A), phytol (B) and geranylgeraniol (C).
Fourth, quantitative results of farnesyl acetate, phytol, and geranylgeraniol should be provided using GC or NMR analysis, not a lethality test.
Response. In the line 116 we report that with the lethality test on brine shrimp we obtained the LD50 of the thirteen compounds isolated from petroleum ether extract of leaves from A. macroprophyllata and with this we performed a preliminary screening of the compounds with the lowest LD50 and with a possible increased antitumor activity. Therefore, P6, P7 and P10 were selected for their characterization and identification.
In the table 6 the quantification was calculated using the percentage of area of each of the compounds in the PEAm.
Additionally, the manuscript was sent to English editing in MDPI English editing services

Reviewer 2 Report
Dear Authors
Special thanks for your point to point response
Good Luck
Author Response
Dear reviewer, we appreciate your suggestions and comments that helped us improve the manuscript.